# Thermal Stability and Crystallization Processes of Pd_78_Au_4_Si_18_ Thin Films Visualized via In Situ TEM

**DOI:** 10.3390/nano14070635

**Published:** 2024-04-05

**Authors:** Bingjiao Yu, Rui Zhao, Zhen Lu, Hangbo Su, Binye Liang, Bingjie Liu, Chunlan Ma, Yan Zhu, Zian Li

**Affiliations:** 1State Key Laboratory of Featured Metal Materials and Life-Cycle Safety for Composite Structures, School of Physical Science and Technology, Guangxi University, Nanning 530004, China; 2107301168@st.gxu.edu.cn (B.Y.); 2007301124@st.gxu.edu.cn (H.S.); 2207301068@st.gxu.edu.cn (B.L.); 2Institute of Physics, Chinese Academy of Sciences, Beijing 100190, China; zhaorui@iphy.ac.cn (R.Z.); zhenlu@iphy.ac.cn (Z.L.); 3MIIT Key Laboratory of Aerospace Information Materials and Physics, College of Science, Nanjing University of Aeronautics and Astronautics, Nanjing 211106, China; lllllbj@nuaa.edu.cn (B.L.); yzhu@nuaa.edu.cn (Y.Z.); 4Jiangsu Key Laboratory of Micro and Nano Heat Fluid Flow Technology and Energy Application, School of Mathematics and Physics, Suzhou University of Science and Technology, Suzhou 215009, China; wlxmcl@usts.edu.cn

**Keywords:** nanometallic glasses, crystallization processes, in situ TEM

## Abstract

Amorphous alloys or metallic glasses (MGs) thin films have attracted extensive attention in various fields due to their unique functional properties. Here, we use in situ heating transmission electron microscopy (TEM) to investigate the thermal stability and crystallization behavior of Pd-Au-Si thin films prepared by a pulsed laser deposition (PLD) method. Upon heating treatment inside a TEM, we trace the structural changes in the Pd-Au-Si thin films through directly recording high-resolution images and diffraction patterns at different temperatures. TEM observations reveal that the Pd-Au-Si thin films started to nucleate with small crystalline embryos uniformly distributed in the glassy matrix upon approaching the glass transition temperature $T_{g}\;=\;625\;K$, and subsequently, the growth of crystalline nuclei into sub-10 nm Pd-Si nanocrystals commenced. Upon further increasing the temperature to 673K, the thin films transformed to micro-sized patches of stacking-faulty lamellae that further crystallized into ${\mathrm{Pd}}_{9}{\mathrm{Si}}_{2}$ and ${\mathrm{Pd}}_{3}\mathrm{Si}$ intermetallic compounds. Interestingly, with prolonged thermal heating at elevated temperatures, the ${\mathrm{Pd}}_{9}{\mathrm{Si}}_{2}$ transformed to ${\mathrm{Pd}}_{3}\mathrm{Si}$. Simultaneously, the solute Au atoms initially dissolved in glassy alloys and eventually precipitated out of the ${\mathrm{Pd}}_{9}{\mathrm{Si}}_{2}$ and ${\mathrm{Pd}}_{3}\mathrm{Si}$ intermetallics, forming nearly spherical Au nanocrystals. Our TEM results reveal the unique thermal stability and crystallization processes of the PLD-prepared Pd-Au-Si thin films as well as demonstrate a possibility of producing a large quantity of pure nanocrystals out of amorphous solids for various applications.

## 1. Introduction

Amorphous alloys or metallic glasses (MGs) represent a unique class of materials that possess an overall glassy atomic structure free of crystalline grains and lattice defects [1]. The non-crystalline nature makes them exhibiting superior mechanical or functional properties, such as exceptionally high strength, hardness, wear and corrosion resistance, and thermoplastic processability [2,3,4]. In addition to the fast development in bulk MGs, recent developments in this field also witness a rapid progress of nanoscale MGs [5] in the form of thin films or nanowires [6] or nanoparticles (NPs) [7]. However, nanoscale MGs are metastable materials being prone to crystallization when subjected to thermal treatments. To retain their glassy structures and outstanding properties, it is imperative to prevent crystallization during the heat-involved processing of nanoscale MGs at elevated temperatures. Therefore, understanding thermal stability and crystallization processes in nanoscale MGs is of critical importance for both the fundamental and applied research. From a different perspective, nanoscale MGs could also be used as precursor materials for producing crystalline nano-sized alloys, which also requires an in-depth understanding of crystallization processes of nanoscale MGs. For examples, the free surface of a nanoscale MG acts as the heterogeneous nucleation site for crystallization, because the surface atoms are in higher energy states than the interior atoms [7,8].

Previous experiments employing various forms of broad-beam-based (light or X-ray or neutron) techniques provide macroscopically averaged information about the bulk metallic glasses [9]. However, the finite dimensions and limited volume of nanoscale metallic glasses present great challenges in probing their thermal stability and crystallization processes. In contrast, in situ transmission electron microscopy (TEM) techniques can simultaneously activate and visualize structural changes, and they have been used to observe the dynamic structural changes of nanomaterials, such as the growth of nanocrystals [10] and nanowires [11], and phase transformations in glassy materials [12,13], providing a near atomic understanding of their underlying mechanisms. Recent in situ heating TEM experiments have been used to study the crystallization behavior of MG nanorods and have revealed the finite-size effect on the apparent onset of crystallization [6]. Moreover, the viscous flow of amorphous materials and activation effects caused by electron irradiation in TEM were reported previously [14].

In this work, we use in situ heating TEM to investigate the thermal stability and crystallization processes of Pd-Au-Si thin films prepared by pulsed laser deposition, particularly focusing on the effect of Au substitution on Pd in the eutectic composition Pd-Si thin films. The crystallization kinetics of eutectic ${\mathrm{Pd}}_{82}{\mathrm{Si}}_{18}$ and its Au-substituted ${\mathrm{Pd}}_{78}{\mathrm{Au}}_{4}{\mathrm{Si}}_{18}$ bulks have been thoroughly investigated, but these are mainly focusing on the temperature range around the glass transition temperature [15,16,17]. With new development in thin film preparation, various amorphous solids or metallic glasses can be prepared in the form of thin films or further reduced dimensions [5,18,19]. Therefore, studies of the thermal stability and crystallization processes of thin films become important yet challenging topics because conventional methods suitable for bulks are difficult to apply on thin films. Here, we employ in situ heating TEM experiments to investigate their thermal stability and crystallization behaviors. We aim to reveal the effects of thin film and the Au substitution on the crystallization of the PLD-prepared eutectic ${\mathrm{Pd}}_{78}{\mathrm{Au}}_{4}{\mathrm{Si}}_{18}$ thin films. We first trace the crystal nucleation and growth in thin films at moderate temperatures around the glass transition temperature $T_{g}$. We then raise the specimen temperature well above the $T_{g}$ to observe the crystal phase transformation and Au precipitation in the ${\mathrm{Pd}}_{78}{\mathrm{Au}}_{4}{\mathrm{Si}}_{18}$ system. The observed precipitation of Au nanocrystals out of PLD-prepared ${\mathrm{Pd}}_{78}{\mathrm{Au}}_{4}{\mathrm{Si}}_{18}$ thin films may exemplify an efficient route to prepare pure nanocrystals with a large quantity for various applications.

## 2. Materials and Methods

Pd-Au-Si metallic glass thin films were fabricated by a pulsed laser deposition (PLD) method. The thin films were directly deposited on rotating NaCl substrates inside a vacuum chamber with a base pressure of $2\times 10^{-5}$ Pa. The target of composition ${\mathrm{Pd}}_{85.5}{\mathrm{Au}}_{3}{\mathrm{Si}}_{11.5}$ (in atomic percent) was ablated by excimer KrF laser pulse (λ = 248 nm, t = 25 ns) at a laser repetition of 8 Hz. Prior to deposition, the target was ablated for 4 min. The film thickness (ranging from 10 to 1000 nm) can be finely controlled by regulating the duration time of deposition. The as-deposited thin films were detached from NaCl substrates by dissolving the NaCl in deionized water. The free-standing films were then transported to 1000-mesh Ni-grids for TEM characterization. TEM experiments were conducted via two electron microscopes (Environmental Titan TEM G2 80-300 working at 300 kV, and Talos F200X G2 working at 200 kV, and both from Scientific Fischer Inc., Hampton, NH, USA. High-angle annular dark-field (HAADF) imaging and energy dispersive X-ray spectroscopy (EDXS) measurements were conducted in both microscopes. The use of a heating holder (model 652.IN, Gatan Inc., Pleasanton, CA, USA) allows the specimen temperature to be varied from room temperature up to 1000 K. In the heating TEM experiments, the heating rate is set at about $2\;K\;\min^{-1}$, and the specimens are placed under the vacuum of about $2\times 10^{-5}\;\mathrm{Pa}$ via an electron microscope.

Calculations of binding energies for ${\mathrm{Pd}}_{9}{\mathrm{Si}}_{2}$ and ${\mathrm{Pd}}_{3}\mathrm{Si}$ intermetallics were carried out with self-consistent density functional theory (DFT) in the Vienna Ab initio Simulation package (VASP) [20,21,22]. The Perdew–Burke–Ernzerhof (PBE) functional of the generalized gradient approximation (GGA) is imposed for the exchange correlation functional [23]. The plane wave cutoff energy is chosen as 350 eV, and the convergence condition for energy and force are 1 × 10^−6^ eV, and 1 × 10^−2^ eV/Å, respectively. In optimizing the structures, the selected k-points for sampling the Brillouin zone are 7 × 5 × 7 and 7 × 9 × 7 for ${\mathrm{Pd}}_{3}\mathrm{Si}$ and ${\mathrm{Pd}}_{9}{\mathrm{Si}}_{2}$, respectively.

## 3. Results

### 3.1. Structural Characterization of As-Deposited Pd_78_Au_4_Si_18_ Film

The structural and the compositional characteristics of the as-deposited ${\mathrm{Pd}}_{78}{\mathrm{Au}}_{4}{\mathrm{Si}}_{18}$ films were characterized through TEM measurements. Figure 1a,b display a representative bright-field TEM image and the corresponding selected area electron diffraction (SAED) pattern of ${\mathrm{Pd}}_{78}{\mathrm{Au}}_{4}{\mathrm{Si}}_{18}$ thin film taken from a circular area of a diameter about 800 nm. In this sub-micron scale, the overall amorphous nature of ${\mathrm{Pd}}_{78}{\mathrm{Au}}_{4}{\mathrm{Si}}_{18}$ thin films is confirmed. In sub-nanoscale, high-resolution TEM (HRTEM) and nanobeam electron diffraction (NBED) measurements were conducted. Figure 1c presents a typical HRTEM image, in which a long-range lattice is absent while lattice patterns (marked by circles) of a few nanometers are observed, which are indicative of short-range crystalline ordering within the thin film. The presence of quenched-in crystalline nuclei is further confirmed by the weak reflections in the NBED pattern [24,25], as shown in Figure 1d. Prior to in situ heating TEM experiments, the overall amorphous while locally short-range crystalline nature of ${\mathrm{Pd}}_{78}{\mathrm{Au}}_{4}{\mathrm{Si}}_{18}$ free-standing thin films is established. 

### 3.2. Crystallization Processes of Pd_78_Au_4_Si_18_ Films via In Situ TEM

We conducted in situ heating TEM experiments on the ${\mathrm{Pd}}_{78}{\mathrm{Au}}_{4}{\mathrm{Si}}_{18}$ thin films supported on a Ni grid. Figure 2 shows a typical series of TEM images recording the process of heating up the specimen from room temperature (Figure 2a) to about 1000 K. Upon reaching 653 K (380 °C), very fine crystallites began to nucleate out of the glassy film, as seen in Figure 2b. It is worth noting that the nucleation process is kinetically rather slow when the specimen temperature was kept below 653 K. In other words, neither the number of crystalline nuclei nor their sizes exhibit substantial change over a few hours. 

With further increasing specimen temperature, one can observe a rapid growth of sub-10-nm crystallites (Figure 2c at 663 K) and the emergence of sub-micron-size patches of tweed-like contrast (Figure 2d at 673 K). These tweed-like patches are non-stoichiometric crystalline ${\mathrm{PdSi}}_{x}$ with a high density of stacking faults, which will transform to stoichiometric ${\mathrm{Pd}}_{9}{\mathrm{Si}}_{2}$ or ${\mathrm{Pd}}_{3}\mathrm{Si}$ phases upon further increasing the specimen temperature, as seen in Figure 2e–h with indicated temperatures. Note that the thermally-driven transformation process of tweed-like patches to crystalline ${\mathrm{Pd}}_{9}{\mathrm{Si}}_{2}$ and ${\mathrm{Pd}}_{3}\mathrm{Si}$ phases will be thoroughly examined below in Figure 3 and Figure 4, respectively. Upon increasing the specimen temperature above 973 K, the tweed-like structure completely turned to highly crystalline ${\mathrm{Pd}}_{9}{\mathrm{Si}}_{2}$ or ${\mathrm{Pd}}_{3}\mathrm{Si}$ phases [26,27,28], as seen in Figure 2i. Interestingly, Figure 2i also shows the presence of 100 nm particles that are proved to be pure Au precipitating from the original Pd(Au)-Si crystalline film, which will be thoroughly described below.The whole process of thermal heating was recorded in Appendix A.

Despite the fact that all of the crystallization processes can be visualized at the nanoscale via in situ TEM and the exact determination of the resultant crystalline phases, some key parameters regarding the crystallization kinetics [29,30,31] are difficult to estimate, which is in part due to the lack of fine control over the heating rate. Future studies of in situ TEM techniques with improved fine control over the heating rate will address these difficulties. Instead, we emphasize the direct observation of crystallization processes via in situ TEM and the accurate determination of resultant Pd-Si phases as well as the precipitation of pure Au nanocrystals.

### 3.3. Transformation of Metastable Pd-Si to Stable Pd_9_Si_2_ and Pd_3_Si

We proceed to examine the tweed-like metastable ${\mathrm{PdSi}}_{x}$ phase with high-density planar defects and their transformation to stoichiometric ${\mathrm{Pd}}_{9}{\mathrm{Si}}_{2}$ and ${\mathrm{Pd}}_{3}\mathrm{Si}$ single-crystal phases upon further thermal heating. Figure 3a shows a distinct grain boundary separating two regions marked by A and B. In region A of the ${\mathrm{PdSi}}_{x}$ phase, Figure 3b depicts a high-resolution TEM lattice image with high-density stacking faults, which cause the streaking reflections in the corresponding SAED pattern in Figure 3c. Note that it remains a challenge to determine the exact crystal structure and chemical composition of the metastable ${\mathrm{PdSi}}_{x}$ phase because of its metastable and faulty lattice. Previous studies reported that the metastable ${\mathrm{PdSi}}_{x}$ structure could be either monoclinic or triclinic unit cells [17,32,33]. These stacking faults are of higher energy and will tend to transform into a lower energy state upon prolonged thermal treatment. Then, the transition can occur from a stacking fault crystal (region A in Figure 3a) toward a single crystal structure (region B in Figure 3a) along the direction indicated by the white arrows, which corresponds to the (020) crystal plane in the SAED pattern along the $\left[1\;0\;\bar{1}\right]$ zone axis. (Figure 3e). It indicates that ${\mathrm{Pd}}_{9}{\mathrm{Si}}_{2}$ grows along the low-order [010] direction by the nucleation and propagation of unit-cell ledges on the (010) planes.

In region B of the ${\mathrm{Pd}}_{9}{\mathrm{Si}}_{2}$ single phase, Figure 3d,e represent the $\left[1\;0\;\bar{1}\right]$-oriented high-resolution TEM lattice image and their corresponding SAED pattern. The growth direction of ${\mathrm{Pd}}_{3}\mathrm{Si}$ was also determined to be [010], initiating and propagating on the (010) planes [34]. Figure 4a,d represent the metastable ${\mathrm{PdSi}}_{x}$ phase and ${\mathrm{Pd}}_{3}\mathrm{Si}$ single-crystal phase. Figure 4b,c are the high-resolution lattice image and the SAED pattern of the metastable phase, where Figure 4e,f depict the ${\mathrm{Pd}}_{3}\mathrm{Si}$ single-crystal phase. Note that the ${\mathrm{Pd}}_{9}{\mathrm{Si}}_{2}$ and ${\mathrm{Pd}}_{3}\mathrm{Si}$ intermetallics were also previously reported to be stable down to room temperature [28,35].

### 3.4. Growth of Pd_3_Si at the Expense of Pd_9_Si_2_


Figure 5a shows a typical TEM image of a mixture of ${\mathrm{Pd}}_{9}{\mathrm{Si}}_{2}$ and ${\mathrm{Pd}}_{3}\mathrm{Si}$ crystalline phases. Analyses of SAED patterns allow one to identify the particular structure for each grain, as shown in Figure 5a, where 9-2 marks the ${\mathrm{Pd}}_{3}\mathrm{Si}$ grain, 3-1 marks the ${\mathrm{Pd}}_{3}\mathrm{Si}$ grain, and yellow-dashed lines mark the grain boundaries. The areas in ${\mathrm{Pd}}_{9}{\mathrm{Si}}_{2}$ grains denoted by letters A and B were selected for recording SAED patterns, as shown in Figure 5b,c, respectively. The sharp and bright reflections in both SAED patterns suggest the single crystal nature of the ${\mathrm{Pd}}_{9}{\mathrm{Si}}_{2}$ grains. 

With prolonged isothermal heat treatment, we observed a phase transformation of ${\mathrm{Pd}}_{9}{\mathrm{Si}}_{2}$ into ${\mathrm{Pd}}_{3}\mathrm{Si}$: namely, a continuous growth of ${\mathrm{Pd}}_{3}\mathrm{Si}$ at the expense of ${\mathrm{Pd}}_{9}{\mathrm{Si}}_{2}$. To visualize directly the ${\mathrm{Pd}}_{9}{\mathrm{Si}}_{2}$ to ${\mathrm{Pd}}_{3}\mathrm{Si}$ transformation, we track the same area by recording TEM images during isothermal heating. Figure 5d shows the same area as marked in Figure 5a, and the regions marked by C and D correspond to A and B in Figure 5a. The SAED patterns for regions C and D are shown in Figure 5e,f, respectively. These SAED patterns can be indexed with the ${\mathrm{Pd}}_{3}\mathrm{Si}$ structure, demonstrating that the original ${\mathrm{Pd}}_{9}{\mathrm{Si}}_{2}$ grains have transformed to ${\mathrm{Pd}}_{3}\mathrm{Si}$ grains. The growth directions of continuous ${\mathrm{Pd}}_{3}\mathrm{Si}$ grains at the expense of ${\mathrm{Pd}}_{9}{\mathrm{Si}}_{2}$ ones are indicated by white arrows in Figure 5a. The detailed transformation process is given in the Appendix A.

### 3.5. Precipitation of Au NPs at Elevated Temperatures

In addition to the formation of micron-sized ${\mathrm{Pd}}_{3}\mathrm{Si}$ grains, near-spherical-shaped NPs are observed in Figure 5a,d during prolonged isothermal heating above 873 K (or 600 °C). We carried out HAADF imaging and EDXS mapping to characterize these NPs. Figure 6a shows a typical HAADF image of NPs distributed in the ${\mathrm{Pd}}_{3}\mathrm{Si}$ grains. Since HAADF intensity scales with the atomic number Z of constituent elements, the brighter contrast of NPs can be attributed to the high atomic number of Au (Z = 79). Figure 6b,c represent the NBED pattern and the HRTEM lattice image of individual Au NPs, indicating their highly crystalline phase. Figure 6d–f show the EDXS mapping for the constituent elements of Pd, Si, and Au, respectively, which unambiguously reveal the precipitation of Au NPs out of the ${\mathrm{Pd}}_{78}{\mathrm{Au}}_{4}{\mathrm{Si}}_{18}$ thin films upon thermal heat treatment. The EDXS maps of Au (Figure 6f) were used to calculate the Au nanoparticle size distribution, as shown in Figure 6g. This size distribution follows a log-normal distribution with a mean value of 124 ± 5 nm. Note that some of the Au NPs in Figure 6f exhibit elongated shapes rather than spherical ones because of their coalescence and Ostwald ripening in prolonged thermal treatment. With fine control over the thermal treatment temperature and duration time, we can obtain a nearly spherical shape of Au NPs well dispersed in the ${\mathrm{Pd}}_{3}\mathrm{Si}$ single-crystal thin films.

To understand the Au precipitating out of Pd-Si thin film, we check the mixing enthalpy for a ternary compound system. According to the No. 104 entry of Table 1 in the work [36] presented by Takeuchi and Inoue, the mixing enthalpy $\Delta H^{\mathrm{mix}}$ in unit of $\left(\mathrm{kJ}\cdot {\mathrm{mol}}^{-1}\right)$ values of Au-Pd-Si subsystems calculated based on Miedema’s macroscopic model are 0 for Au-Pd, -30 for Au-Si, and -55 for Pd-Si. Since the Au content is a minor part in the eutectic alloy of ${\mathrm{Pd}}_{78}{\mathrm{Au}}_{4}{\mathrm{Si}}_{18}$, upon heating, it tends to crystallize into ${\mathrm{Pd}}_{9}{\mathrm{Si}}_{2}$ and ${\mathrm{Pd}}_{3}\mathrm{Si}$, and consequently, the solute Au elements tend to precipitate to form nanocrystals at elevated temperatures.

## 4. Discussion

### 4.1. Effects of Quenched-in Nuclei on Crystallization Processes of Pd-Au-Si Thin Films

It is now well-documented [17,37,38] that in metal–metalloid amorphous alloys, there exist short- and medium-range-ordered crystallites with rich atomistic motifs depending on many intrinsic (composition) or extrinsic (fabrication and processing) factors. These quenched-in ultrafine crystallites play critical roles in the nucleation and subsequent growth of crystals out of amorphous alloys. In this work, we also detected the presence of nanoscale crystallites uniformly distributed in the PLD-deposited Pd-Au-Si thin films, and we observed the incipient crystallization commencing around the quenched-in crystallites at relatively low temperatures (see Figure 1). The presence of quenched-in crystallites also explains partially the lower transition and initial crystallization temperatures $T_{g}$ and $T_{x}$ in comparison to the arc-melt bulk MGs and melt-spun ribbons [16,28,39,40].

The lamellar structures observed in our PLD-prepared thin films have been reported in previous TEM experiments. For example, Masumoto et al. [41] first reported that at the beginning of crystallization, the MS-I phase precipitated out from the amorphous matrix. Upon increasing the annealing temperature, the MS-I phase then transformed to the MS-II phase, which finally reached the equilibrium ${\mathrm{Pd}}_{3}\mathrm{Si}$ phase. According to Masumoto et al. [41], MS-I is a Pd-Si solid solution which has a FCC unit cell with a = 0.40 nm, and MS-II is an unknown complex superlattice structure. Duhaj et al. [42] reported that the complex appearance of MS-II might be attributed to the morphology of mixed ${\mathrm{Pd}}_{9}{\mathrm{Si}}_{2}$ and ${\mathrm{Pd}}_{3}\mathrm{Si}$ phases at various temperatures. Brearley et al. [34] investigated the lamellar structure of mixed ${\mathrm{Pd}}_{9}{\mathrm{Si}}_{2}$ and ${\mathrm{Pd}}_{3}\mathrm{Si}$ phases. Wu et al. [28] claimed that the morphologically complex MS-II phase can be formed in the whole amorphous formation region in the Pd-Si system displaying various crystallization processes involving transition states and phase transformation. These lamellar structures are also observed in our TEM experiments on the PLD-prepared Pd-Au-Si thin films and exhibit many similar crystallization behaviors and phenomena.

### 4.2. Transformation of Pd_9_Si_2_ to Pd_3_Si at Elevated Temperatures

To understand the ${\mathrm{Pd}}_{9}{\mathrm{Si}}_{2}$ to ${\mathrm{Pd}}_{3}\mathrm{Si}$ transformation above 923K in the PLD-prepared Pd-Au-Si thin films, we consider their atomic structures, binding energies and thermodynamic properties. Both ${\mathrm{Pd}}_{9}{\mathrm{Si}}_{2}$ and ${\mathrm{Pd}}_{3}\mathrm{Si}$ share a common orthorhombic atomic structure with a space group Pnma [27,43]. Note that structurally, the ${\mathrm{Pd}}_{9}{\mathrm{Si}}_{2}$ phase is somewhat related to ${\mathrm{Pd}}_{3}\mathrm{Si}$; namely, it is a superlattice which contains four times as many Pd atoms (48 total) as ${\mathrm{Pd}}_{3}\mathrm{Si}$ with the Si atoms randomly occupying only 11 of the 16 possible z = 1/4 and 3/4 sites [27]. The metal–metalloid bonding is considerably more isotropic than that in ${\mathrm{Pd}}_{3}\mathrm{Si}$ [34]. We performed DFT calculations using a VASP software on the structural optimization and binding energy evaluation of both ${\mathrm{Pd}}_{9}{\mathrm{Si}}_{2}$ and ${\mathrm{Pd}}_{3}\mathrm{Si}$ (calculation details are given in the Materials and Methods section). Starting from experimental crystal structures, we change the scaling coefficient to perform a series of structural relaxations. Selecting the result with the lowest energy, subtle adjustments based on the lattice constant in the minimum direction are further carried out to identify the lowest-energy structure. The calculated lattice parameters for both phases at equilibrium are listed in Table 1.

With the structure optimization results, one can calculate the binding energy as $E_{B}=E_{{\mathrm{Pd}}_{n}{\mathrm{Si}}_{m}}-E_{{\mathrm{Pd}}_{n}}-E_{{\mathrm{Si}}_{m}}$. The total binding energy of the ${\mathrm{Pd}}_{n}{\mathrm{Si}}_{m}$ structure is $E_{{\mathrm{Pd}}_{n}{\mathrm{Si}}_{m}}$, and $E_{{\mathrm{Pd}}_{n}}$ and $E_{{\mathrm{Si}}_{m}}$ are the energies in their respective equilibrium bulk phases. The resultant binding energies are summarized in Table 1. Since their compositions are very different, it is informative to compare the binding energies per unit volume $E_{\mathrm{BV}}=\frac{E_{B}}{V}$, as tabulated in Table 1. The binding energy for the ${\mathrm{Pd}}_{3}\mathrm{Si}$ phase is 10.8meV Å^−3^ lower than that of the ${\mathrm{Pd}}_{9}{\mathrm{Si}}_{2}$ phase, suggesting that ${\mathrm{Pd}}_{3}\mathrm{Si}$ is energetically stable over the ${\mathrm{Pd}}_{9}{\mathrm{Si}}_{2}$ phase at zero-temperature.

It is worth noting the melting temperatures $\left(T_{m}\right)$ for ${\mathrm{Pd}}_{3}\mathrm{Si}$ of 1318 K and ${\mathrm{Pd}}_{9}{\mathrm{Si}}_{2}$ of 1096 K, suggesting their inherent thermal stability [34]. Here, our TEM experiments reveal that the ${\mathrm{Pd}}_{9}{\mathrm{Si}}_{2}$ to ${\mathrm{Pd}}_{3}\mathrm{Si}$ transformation occurred around 823 K, which is significantly lower compared to $T_{m}=1096\;K$ of bulk ${\mathrm{Pd}}_{9}{\mathrm{Si}}_{2}$. The large portion of atoms at the surfaces of thin films may help reduce the melting temperature.

### 4.3. Precipitation of Au NPs from Crystallization of Amorphous Pd-Au-Si System

During the formation of ${\mathrm{Pd}}_{9}{\mathrm{Si}}_{2}$ and ${\mathrm{Pd}}_{3}\mathrm{Si}$ intermetallics in the ${\mathrm{Pd}}_{78}{\mathrm{Au}}_{4}{\mathrm{Si}}_{18}$ thin films at high temperatures above 923 K, the Au atoms simultaneously precipitated out of the Pd-Si crystals to form Au NPs. These Au NPs uniformly dispersed in the self-standing thin films upon the completion of ${\mathrm{Pd}}_{3}\mathrm{Si}$ crystallization. When the specimen temperature further increases, the isolated Au NPs become mobile and tend to coalesce into larger NPs. To understand the precipitation phenomenon of Au out of the ternary Pd-Au-Si thin films at evaluated temperature, it is informative to check their binary Au-Pd, Au-Si and Pd-Si phase diagrams. In the Au-Pd phase diagram, these elements form solid solution across the whole composition range. By contrast, there exits a limited solubility in Au-Si phase diagram, and there exist several intermetallic compounds including ${\mathrm{Pd}}_{5}\mathrm{Si}$, ${\mathrm{Pd}}_{9}{\mathrm{Si}}_{2}$, ${\mathrm{Pd}}_{3}\mathrm{Si}$ and ${\mathrm{Pd}}_{2}\mathrm{Si}$ in the Pd-Si phase diagram [44,45]. In the as-prepared ${\mathrm{Pd}}_{78}{\mathrm{Au}}_{4}{\mathrm{Si}}_{18}$ system, the content of Au is about 4 at.% as minor solute elements. At elevated temperature, the Pd-Si crystallized into ${\mathrm{Pd}}_{9}{\mathrm{Si}}_{2}$ and ${\mathrm{Pd}}_{3}\mathrm{Si}$ intermetallics, while the small amount of solute Au elements precipitated into crystalline NPs. It is worth noting that in our TEM experiments, the ${\mathrm{Pd}}_{78}{\mathrm{Au}}_{4}{\mathrm{Si}}_{18}$ thin films are kept in a vacuum of $2\times 10^{-5}\;\mathrm{Pa}$, and in the magnetic field, they are kept in a vacuum of around 2T.

It remains an open question regarding the nanoscale effect (finite thickness) of thin films on the crystallization processes as well as the influence of specific TEM vacuum and magnetic-field conditions during heating treatment. Previous reports suggest that the sublimation of metals or alloys can occur in TEM heating experiments at temperatures significantly lower than their respective melting temperatures [46,47].

Our survey of literature suggests that this study provides the first report of the precipitation of Au NPs out of the Pd-Au-Si ternary systems. Previous investigations of the ${\mathrm{Pd}}_{82-x}{\mathrm{Au}}_{x}{\mathrm{Si}}_{18}$ ternary system largely focused on the crystallization processes below or close to $T_{g}$ and revealed a rich thermal behaviors of Pd-Au-Si melt-spun ribbon or arc-melt bulk [16,48,49]. Here, our in situ heating TEM experiments allow the specimen temperature to reach around 1000 K, enabling the observation of the precipitation of Au out of Pd-Au-Si amorphous thin films around such elevated temperatures. This precipitation mechanism in amorphous thin films could be further extended to prepare pure noble metal nanoparticles or clusters with a large quantity for a range of applications.

## 5. Conclusions

In summary, we used a pulsed laser deposition method to prepare Pd-Au-Si amorphous thin films on NaCl substrates and obtained free-standing thin films supported on the Ni-grid by dissolving the NaCl substrates. Prior to in situ heating TEM experiments, the structural and compositional characteristics of the thin films were carefully checked, revealing the compositional ${\mathrm{Pd}}_{78}{\mathrm{Au}}_{4}{\mathrm{Si}}_{18}$ with the presence of quenched-in nanoscale crystallites. Thermal TEM experiments were then conducted to investigate the thermal evolution and crystallization processes of the ${\mathrm{Pd}}_{78}{\mathrm{Au}}_{4}{\mathrm{Si}}_{18}$ thin films. Upon heating at various temperatures, TEM observations reveal that the incipient crystallization initiated at around 653±5K, and the 1–3 nm crystallites grew up to sub-10 nm upon reaching 663K. As the specimen temperature reached 673K, sub-micron lamellae structures emerged and rapidly engulfed the pre-formed crystallites until the temperature reached 723K. As specimen temperature increased to 823K, the metastable lamellar structures transformed to ${\mathrm{Pd}}_{9}{\mathrm{Si}}_{2}$ and ${\mathrm{Pd}}_{3}\mathrm{Si}$ intermetallics. Furthering the specimen temperature to 823K, the ${\mathrm{Pd}}_{3}\mathrm{Si}$ crystals continued growth at the expense of ${\mathrm{Pd}}_{9}{\mathrm{Si}}_{2}$. Around such elevated temperatures, the Au nanocrystals simultaneously precipitated out of the Pd-Au-Si solid solution.

The ${\mathrm{Pd}}_{78}{\mathrm{Au}}_{4}{\mathrm{Si}}_{18}$ amorphous alloy undergoes multiple steps of atomic diffusion and rearrangement during thermal heating, gradually transitioning from an amorphous state to crystalline ${\mathrm{Pd}}_{3}\mathrm{Si}$ and ${\mathrm{Pd}}_{9}{\mathrm{Si}}_{2}$ intermetallic compounds and precipitating pure Au NPs at higher temperatures. It is worth noting that at higher temperatures, ${\mathrm{Pd}}_{9}{\mathrm{Si}}_{2}$ transformed into the ${\mathrm{Pd}}_{3}\mathrm{Si}$ intermetallic phase in the PLD-prepared ${\mathrm{Pd}}_{78}{\mathrm{Au}}_{4}{\mathrm{Si}}_{18}$ thin films, which was not reported in previous studies. We attribute the Au precipitation phenomenon to the large negative enthalpy of Pd and Si to form intermetallics, while the Pd-Au or Au-Si do have relatively small negative enthalpy; in turn, pure Au precipitates out of the Pd-Si system. This effect can be further exploited to prepare pure single-element metal or alloy nanocrystals uniformly dispersed on supporting amorphous or crystalline solids.

## Figures and Tables

**Figure 1 nanomaterials-14-00635-f001:**
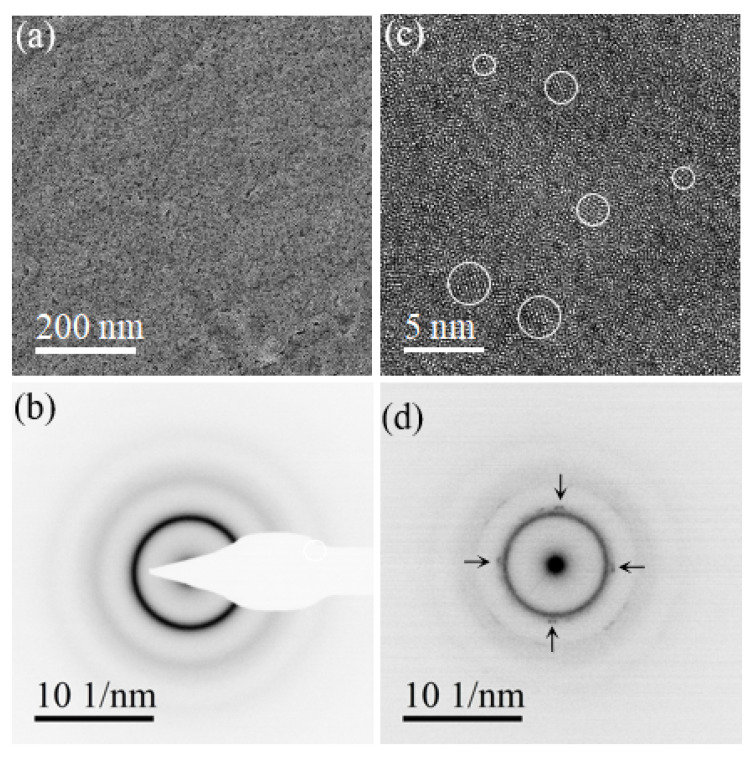
TEM characterization of amorphous ${\mathrm{Pd}}_{78}{\mathrm{Au}}_{4}{\mathrm{Si}}_{18}$ metallic glass thin films. (**a**) Bright-field TEM image survey and (**b**) typical SAED pattern of metallic glass phase. (**c**) HRTEM image with short-range ordered regions marked by white circles. (**d**) NBED pattern and the weak reflections marked by arrows owing to the presence of short-range ordered crystallites.

**Figure 2 nanomaterials-14-00635-f002:**
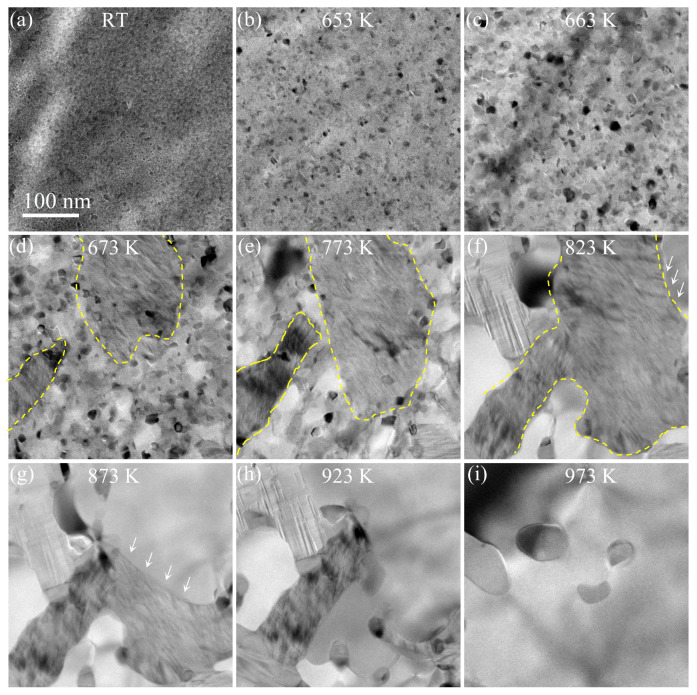
Crystallization processes of ${\mathrm{Pd}}_{78}{\mathrm{Au}}_{4}{\mathrm{Si}}_{18}$ thin films observed via in situ heating TEM. (**a**) Initial amorphous state at room temperature. Early stages of homogeneous nucleation and subsequent growth of nanocrystallites at (**b**) 653 ± 1 K and (**c**) at 663 ± 1 K. (**d**–**f**) are the continuous growth of crystallites into lamella-like stacking faults of ${\mathrm{PdSi}}_{x}$ (denoted as yellow dashed region). (**g**–**i**) With elevated temperature and prolonged heat duration, intermediate phase ${\mathrm{PdSi}}_{x}$ transforms to ${\mathrm{Pd}}_{3}\mathrm{Si}$ and ${\mathrm{Pd}}_{9}{\mathrm{Si}}_{2}$, and finally to ${\mathrm{Pd}}_{3}\mathrm{Si}$, along with the precipitation of Au nanocrystals at 973 K.

**Figure 3 nanomaterials-14-00635-f003:**
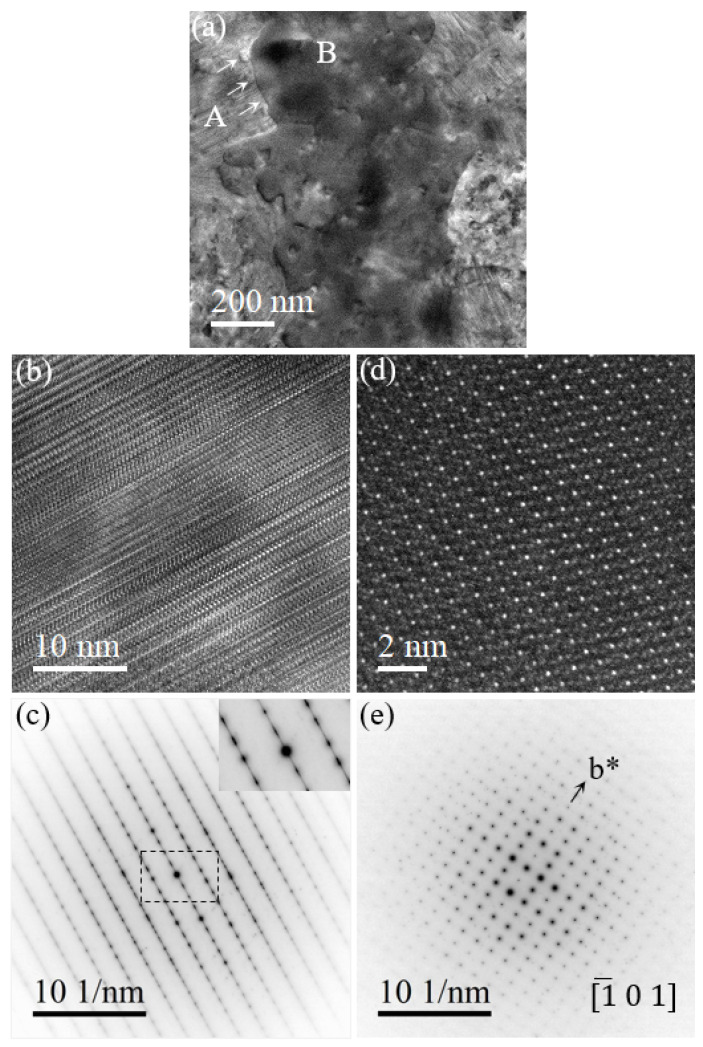
Transformation of metastable ${\mathrm{PdSi}}_{x}$ phase to stable ${\mathrm{Pd}}_{9}{\mathrm{Si}}_{2}$ phase. (**a**) Bright-field TEM image of coexistence of two phases. Letter A marks the stacking-faulted ${\mathrm{PdSi}}_{x}$ phase, letter B marks the ${\mathrm{Pd}}_{9}{\mathrm{Si}}_{2}$ single-crystal phase, and white arrows denote the growth direction of ${\mathrm{Pd}}_{9}{\mathrm{Si}}_{2}$. (**b**) HRTEM lattice image and (**c**) corresponding SAED pattern of lamellae area A. Central part of SAED is shown as inset. (**d**) HRTEM lattice image and (**e**) corresponding SAED pattern of ${\mathrm{Pd}}_{9}{\mathrm{Si}}_{2}$ area B. In (**e**) b* denotes the reciprocal axis.

**Figure 4 nanomaterials-14-00635-f004:**
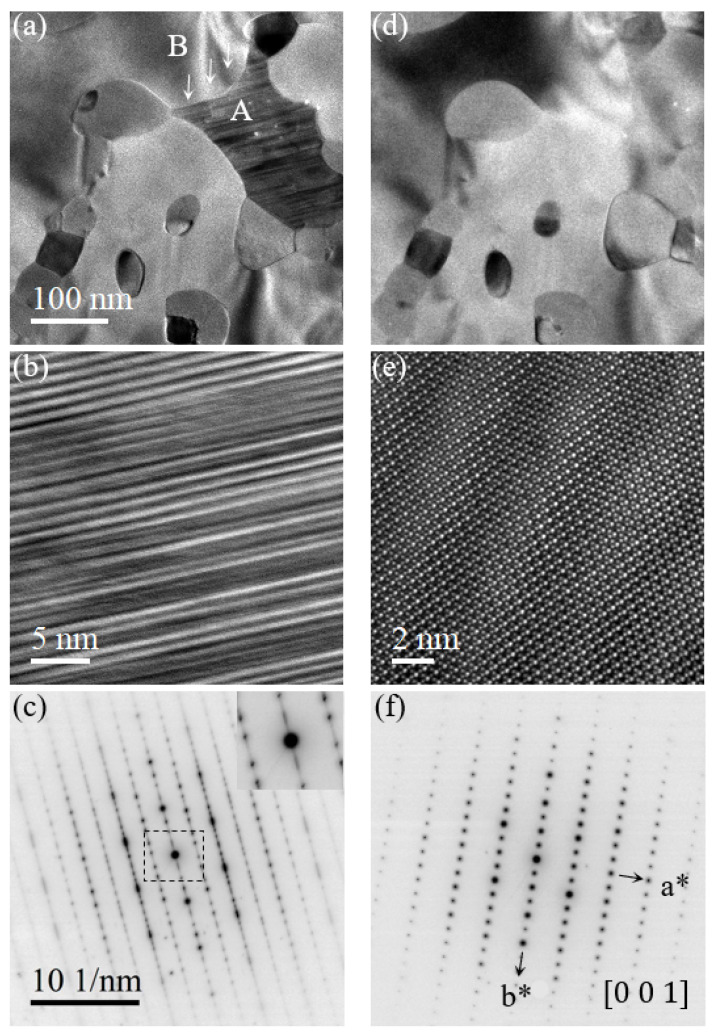
Transformation of metastable ${\mathrm{PdSi}}_{x}$ phase to stable ${\mathrm{Pd}}_{3}\mathrm{Si}$ phase. (**a**) Bright-field TEM image of coexistence of two phases. Letter A marks the stacking-faulted ${\mathrm{PdSi}}_{x}$ phase, letter B marks the ${\mathrm{Pd}}_{3}\mathrm{Si}$ single-crystal phase, and white arrows denote the growth direction of ${\mathrm{Pd}}_{3}\mathrm{Si}$. (**b**,**c**) are the HRTEM lattice images and the corresponding SAED patterns of area A. Central part of SAED is shown as inset. (**d**) Bright-field image of Figure (**a**) completely transformed into the ${\mathrm{Pd}}_{3}\mathrm{Si}$ phase. (**e**,**f**) are the HRTEM lattice images and the corresponding SAED patterns of area B. In (**f**) a* and b* denote the reciprocal axes.

**Figure 5 nanomaterials-14-00635-f005:**
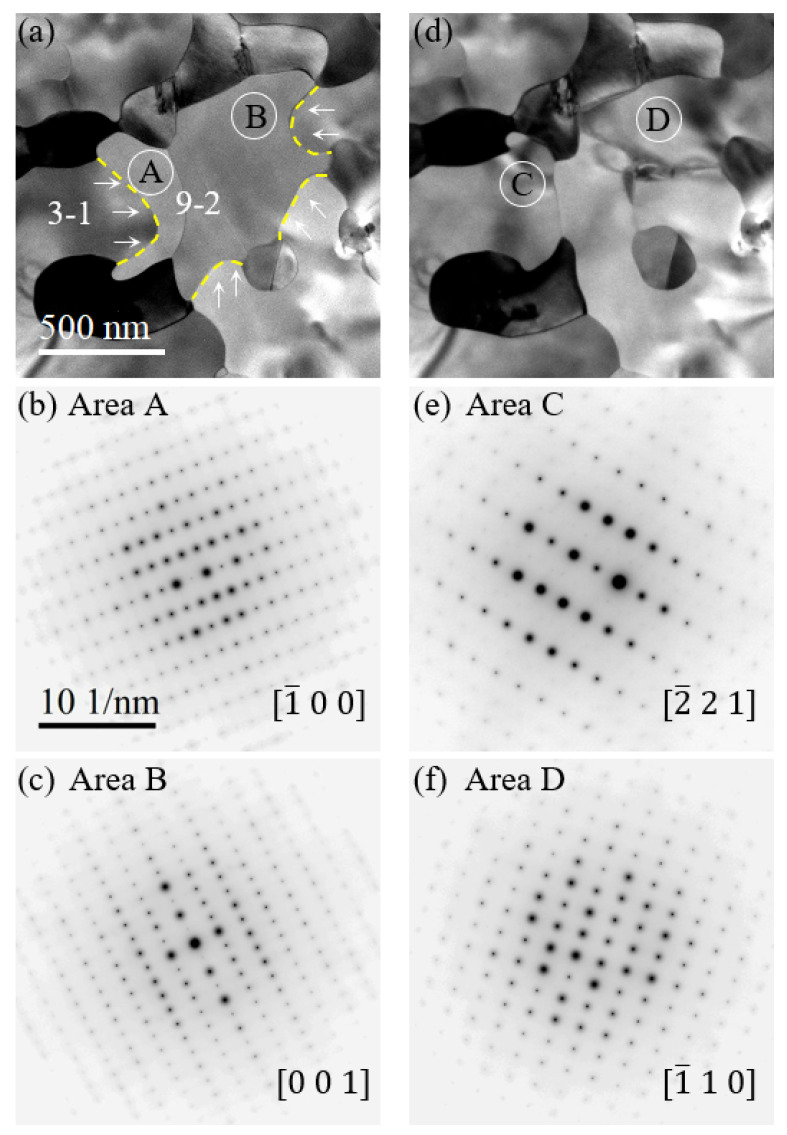
Continuous growth of ${\mathrm{Pd}}_{3}\mathrm{Si}$ at the expense of ${\mathrm{Pd}}_{9}{\mathrm{Si}}_{2}$ during prolonged isothermal heat treatment. (**a**) Bright-field TEM image of coexistence of ${\mathrm{Pd}}_{3}\mathrm{Si}$ marked by 3-1 and ${\mathrm{Pd}}_{9}{\mathrm{Si}}_{2}$ marked by 9-2. Yellow-dash lines denote the phase boundaries, white arrows denote the growth direction of ${\mathrm{Pd}}_{3}\mathrm{Si}$, letters A and B mark the ${\mathrm{Pd}}_{9}{\mathrm{Si}}_{2}$ regions, and the corresponding SAED patterns are shown in (**b**,**c**), respectively. (**d**) Identical area as (**a**) but after the complete growth of ${\mathrm{Pd}}_{3}\mathrm{Si}$. Note that the areas C and D mark the respective areas A and B in (**a**), and their corresponding SAED patterns are shown in (**e**,**f**). Note that (**c**,**e**,**f**) share the same scale bar in (**b**).

**Figure 6 nanomaterials-14-00635-f006:**
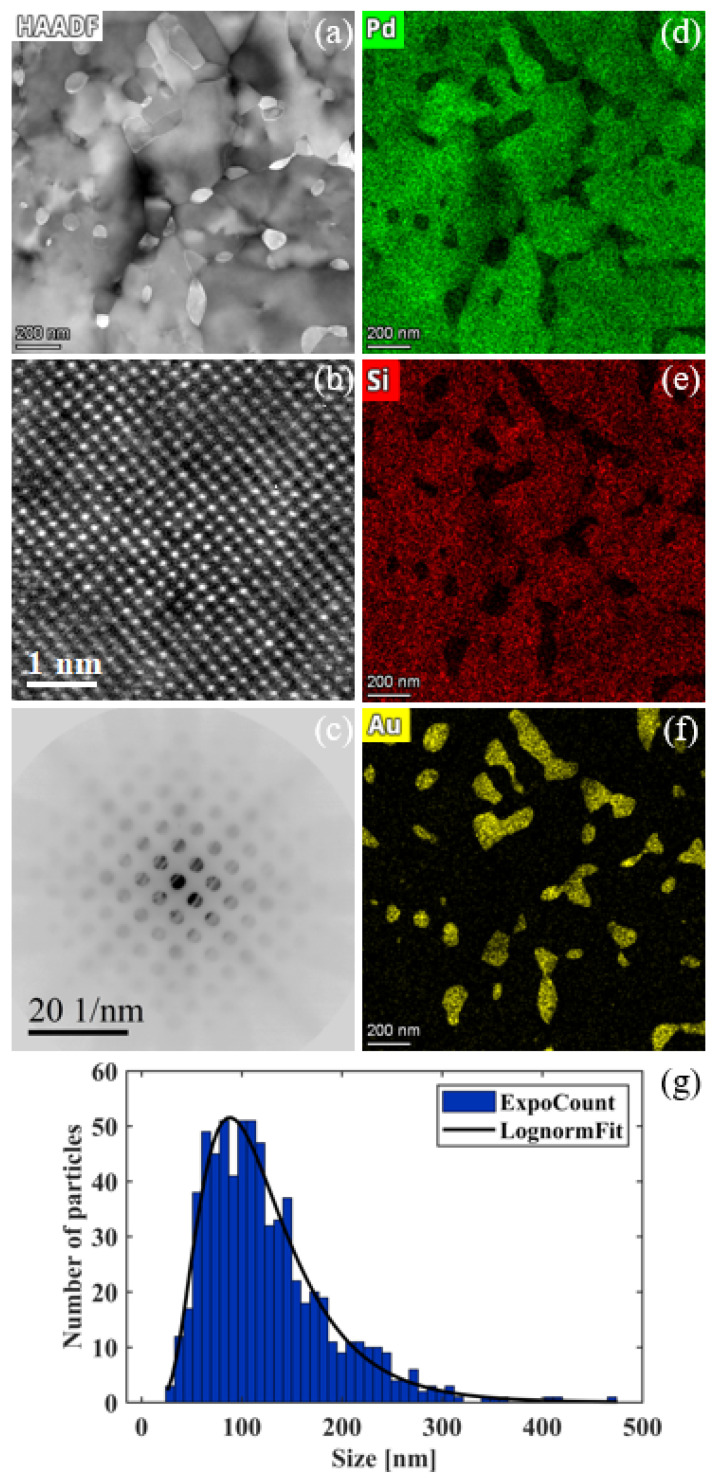
Precipitation of Au NPs characterized via HAADF imaging and EDXS. (**a**) HAADF survey image displays the Au NPs. (**b**) HRTEM lattice image and (**c**) the corresponding nanobeam electron diffraction pattern of Au nanoparticles. EDXS spectroscopic mapping for (**d**) Pd, (**e**) Si, and (**f**) Au elements. (**g**) Fitting of a log-normal function to the size distribution of Au NPs.

**Table 1 nanomaterials-14-00635-t001:** The DFT calculations of lattice constants and binding energies of ${\mathrm{Pd}}_{3}\mathrm{Si}$ and ${\mathrm{Pd}}_{9}{\mathrm{Si}}_{2}$.

	a (Å)	b (Å)	c (Å)	V (Å^3^)	$E_{{\mathrm{Pd}}_{n}{\mathrm{Si}}_{m}}$ (eV)	$E_{B}^{\mathrm{bulk}}$ (eV)	$E_{\mathrm{BV}}^{\mathrm{bulk}}$ (meV/Å^3^)
${\mathrm{Pd}}_{3}\mathrm{Si}$	5.934	7.639	5.350	242.529	−93.6	−9.8	−40.2
${\mathrm{Pd}}_{9}{\mathrm{Si}}_{2}$	9.263	7.616	9.659	681.417	−249.8	−20.0	−29.4

## Data Availability

The data presented in this study are available on request from the corresponding author.

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
