# Peer review of "Thermal Stability and Crystallization Processes of Pd_78_Au_4_Si_18_ Thin Films Visualized via In Situ TEM"

_nanomaterials, 2024, doi:10.3390/nano14070635_

Round 1
Reviewer 1 Report
Comments and Suggestions for Authors
The article "Crystallization kinetics of Pd78Au4Si18 thin films visualized via in situ TEM" is devoted to the study of processes occurring during crystallization of amorphous Pd78Au4Si18 films obtained by magnetron evaporation. The studies were carried out by in situ TEM methods, also electron diffraction methods (SAED, NBD) were used. Heating of thin film samples was carried out directly in the transmission electron microscope column, which allowed to study the crystallization process of the films in sufficient detail.
General remark.
The article leaves a double impression. On the one hand, the studies by transmission electron microscopy methods, including HRTEM, as well as electron diffraction methods, were carried out at a very high level. On the other hand, it remains unclear why the authors call it "a study of crystallization kinetics". The authors do not make any estimates of the kinetic parameters of the crystallization process of the studied Pd78Au4Si18 films in the paper.
The following are specific comments.
Line 1. Abstract. " Nanoscale amorphous alloys or metallic glasses (MGs) have recently attracted extensive attention ..."
First, it is not clear what "Nanoscale amorphous alloys" means, because amorphous structure already implies a size smaller than 1 nm. Second, it is hardly appropriate to use the word "recently" for materials that have attracted attention for more than 50 years.
Lines 1-17. Abstract.
It is not clear what the scientific novelty of this work is. The Abstract contains only experimental results and nothing more.
Lines 17. "clean nanocrystals"
probably "pure nanocrystals" would be better.
Lines 19-56. Introduction.
The Introduction section contains a lot of general words about the importance of the study of amorphous materials and not a word about the relevance of the study of Pd-Au-Si films. It also remains unclear why the authors decided to study this particular composition - Pd78Au4Si18.
Lines 57-86. Materials and Methods.
Line 60. " Target of composition Pd85.5Au3Si11.5"
It is necessary to specify in what units the composition is given - atomic or weight percent.
Lines 63-65. "The thermal properties of the samples were measured using a simultaneous thermal analyzer (STA) (model 8000, Perkin-Elmer, Inc.)."
The authors do not cite any results obtained by the STA method in this paper (indeed, this method is not even mentioned in the paper).
Lines 73-75. "It is worth noting that to carefully observe the crystallization process, the specimen temperature will be finely controlled with a manual mode, instead of auto mode."
It is not clear why the authors used "manual mode". Automatic control provides guaranteed maintenance of both the heating rate and the fixed temperature. It is unlikely that the authors could have done this better in manual mode.
Line 102. 3.2. Crystallization kinetics of Pd78Au4Si18 films via in situ TEM
It is not clear what the authors mean when they write about crystallization kinetics. Neither in this paragraph nor further on in the article are any estimates of the kinetic parameters of the crystallization process of Pd-Au-Si films, as well as the formation of Pd-Si phases. The authors could have made an estimation of the kinetic parameters if they had performed a series of heating with different heating rates as was done in these papers (DOI: 10.3390/nano13222925, 10.3390/ma1523845457, 10.1016/j.jallcom.2021.159474). In this case, the authors could make an estimation of kinetic parameters (apparent activation energy, pre-exponential factor in the Arrhenius equation) using non-isothermal model-free methods. The authors could also estimate kinetic parameters of the crystallization process and phase formation if they investigated their samples by simultaneous thermal analysis (STA). Heating at different rates would have allowed them not only to estimate kinetic parameters from STA data, but also to determine a kinetic model for each process. If the authors had used not only STA data, but also in situ electron diffraction data (at least one heating at a rate coinciding with one of the STA heats), then the authors would have been able, similar to these papers (DOI: 10.3390/nano13222925, 10.3390/ma15238457, 10.1016/j.jallcom.2021.159474) interpret the peaks in the STA curves and understand which peak corresponds to which process, such as the crystallization process or the formation of a particular Pd-Si phase.
Line 268. Conclusions.
A description of the results obtained is given and it remains unclear what conclusions are drawn from these studies and what is the scientific novelty of this work. However, the main claim is that the purpose of the work was to study the crystallization kinetics of Pd78Au4Si18 films, and this was not done in this work.
Comments on the Quality of English LanguageThe English should be check. There is some problems with grammar and style.
Author Response
Thank you for your critical comments and useful suggestions. We revised the manuscript accordingly, please the point-to-point reply to your questions and the marked revision and the revision.

Reviewer 2 Report
Comments and Suggestions for Authors
This is a high-quality work recommended to publish. Authors have used pulsed laser deposition preparing thin amorphous Pd-Au-Si films on NaCl substrate and obtained free-standing thin films supported on Ni-grid by dissolving the substrate. In situ heating technique enabled authors to investigate the thermal stability and crystallization of thin films upon heating inside the TEM where they traced structural changes recording high-resolution images and diffraction patterns at different temperatures. The paper is well-structured providing details to support their observations. Two small amendments are nevertheless needed as follows:
Lines 41-45: Authors shall be aware that apart from structural changes observed the TEM techniques enables visualisation of morphological changes of nano-scale materials including viscous flow of amorphous materials and activation effects caused by electron irradiation of glasses, see e.g. Zheng, K. et.al. Electron-beam-assisted superplastic shaping of nanoscale amorphous silica. Nat. Commun. 2010, 1, 24. https://doi.org/10.1038/ncomms1021.
Lines 191-196: Authors are reporting presence of nanoscale uniformly distributed quenched-in crystallites observing that incipient crystallization commences around them (referring to manuscript Figure 1). Apart from references quoted it shall be added that the presence of quenched-in crystallites is naturally expected within Johnson-Mehl-Avrami-Kolmogorov model Fanfoni, M.; Tomellini, M. The Johnson-Mehl-Avrami-Kolmogorov model: A brief review. Il Nuovo Cimento D 1998, 20, 1171–1182. https://doi.org/10.1007/BF03185527 and recently confirmed for glasses obtained at ultra-high quenching rates about 1012 K/s with clusters made up of unions of elementary cells of the crystal structure seen at lower quenching rates German, E.I. et.al. Structure of Argon Solid Phases Formed from the Liquid State at Different Isobaric Cooling Rates. Appl. Sci. 2024, 14 (3) 1295 https://doi.org/10.3390/app14031295 .
Author Response

(The authors gave the same response as above.)

Round 2
Reviewer 1 Report
Comments and Suggestions for Authors
The authors have responded to almost all comments, the only complaint is with the following sentence:
Lines 141-142. "... too tedious to estimate kinetic parameters for each heating rate."
This is a wrong expression, because kinetic parameters are determined for the crystallization process (or formation of a particular phase) on the basis of experiments involving several heating rates (at least 3 rates, more is better), not for each heating rate. The authors are right that this is indeed a big job, requiring much more effort than a single heating rate study. But such work allows estimation of kinetic parameters.
Comments on the Quality of English LanguageThe English should be checked.
Author Response
The authors have responded to almost all comments, the only complaint is with the following sentence:
Lines 141-142. "... too tedious to estimate kinetic parameters for each heating rate."
Response: In the revision, we delete this part.
This is a wrong expression, because kinetic parameters are determined for the crystallization process (or formation of a particular phase) on the basis of experiments involving several heating rates (at least 3 rates, more is better), not for each heating rate. The authors are right that this is indeed a big job, requiring much more effort than a single heating rate study. But such work allows estimation of kinetic parameters.
Response: We agree with you that kinetic parameters are important and should be carefully designed to obtain. That is well taken for future study with careful experimental plan.